# SAMPLE EFFICIENT QUALITY DIVERSITY FOR NEURAL CONTINUOUS CONTROL

## ABSTRACT

We propose a novel Deep Neuroevolution algorithm, QD-RL, that combines the strengths of off-policy reinforcement learning (RL) algorithms and Quality Diversity (QD) approaches to solve continuous control problems with neural controllers. The QD part contributes structural biases by decoupling the search for diversity from the search for high return, resulting in efficient management of the exploration-exploitation trade-off. The RL part contributes sample efficiency by relying on off-policy gradient-based updates of the agents. More precisely, we train a population of off-policy deep RL agents to simultaneously maximize diversity within the population and the return of each individual agent. QD-RL selects agents interchangeably from a Pareto front or from a Map-Elites grid, resulting in stable and efficient population updates. Our experiments in the ANT-MAZE and ANT-TRAP environments show that QD-RL can solve challenging exploration and control problems with deceptive rewards while being two orders of magnitude more sample efficient than the evolutionary counterpart.

## 1 INTRODUCTION

Natural evolution has the fascinating ability to produce organisms that are all high-performing in their respective niche. Inspired by this ability to produce a tremendous diversity of living systems within one run, Quality-Diversity (QD) is a new family of optimization algorithms that aim at searching for a collection of both diverse and high-performing solutions (Pugh et al., 2016). While classic optimization methods focus on finding a single efficient solution, the role of QD optimization is to cover the range of possible solution types and to return the best solution for each type. This process is sometimes referred to as "illumination" in opposition to optimization, as the goal of these algorithms is to reveal (or illuminate) a search space of interest (Mouret & Clune, 2015).

QD approaches generally build on black-box optimization methods such as evolutionary algorithms to optimize a population of solutions (Cully & Demiris, 2017). These algorithms often rely on random mutations to explore small search spaces but struggle when confronted to higher-dimensional problems. As a result, QD approaches often scale poorly in large and continuous sequential decision problems, where using controllers with many parameters such as deep neural networks is mandatory (Colas et al., 2020). Besides, while evolutionary methods are the most valuable when the policy gradient cannot be applied safely (Cully et al., 2015), in policy search problem that can be formalized as a Markov Decision Process (MDP), Policy Gradient (PG) methods can exploit the analytical structure of neural networks to more efficiently optimize their parameters. Therefore, it makes sense to exploit these properties when the Markov assumption holds and the controller is a neural network.

From the deep reinforcement learning (RL) perspective, the focus on sparse or deceptive rewards led to realize that maximizing diversity independently from rewards might be a good exploration strategy (Lehman & Stanley, 2011a; Colas et al., 2018; Eysenbach et al., 2018). More recently, it was established that if one can define a small *behavior space* or *outcome space* corresponding to what matters to determine success, maximizing diversity in this space might be the optimal strategy to find a sparse reward (Doncieux et al., 2019).

In this work, we are the first to combine QD methods with PG methods. From one side, our aim is to strongly improve the sample efficiency of QD methods to get neural controllers solving continuous action space MDPs. From the other side, it is to strongly improve the exploration capabilities of deep RL methods in the context of sparse rewards or deceptive gradients problems, such as avoid traps

and dead-ends in navigation tasks. We build on off-policy PG methods to propose a new mutation operator that takes into account the Markovian nature of the problem and analytically exploits the known structure of the neural controller. Our QD-RL algorithm falls within the QD framework described by Cully & Demiris (2017) and takes advantage of its powerful exploration capabilities, but also demonstrates remarkable sample efficiency brought by off-policy RL methods. We compare QD-RL to several recent algorithms that also combine a diversity objective and a return maximization method, namely the NS-ES family (Conti et al., 2018) and the ME-ES algorithm (Colas et al., 2020) and show that QD-RL is two orders of magnitude more sample efficient.

## 2 PROBLEM STATEMENT

We consider the general context of a fully observable Markov Decision Problem (MDP) $(\mathcal{S}, \mathcal{A}, \mathcal{T}, \mathcal{R}, \gamma, \rho_0)$ where $\mathcal{S}$ is the state space, $\mathcal{A}$ is the action space, $\mathcal{T} : \mathcal{S} \times \mathcal{A} \rightarrow \mathcal{S}$ is the transition function, $\mathcal{R} : \mathcal{S} \times \mathcal{A} \rightarrow \mathbb{R}$ is the reward function, $\gamma$ is a discount factor and $\rho_0$ is the initial state distribution. We aim to find a set of parameters $\theta$ of a parameterized policy $\pi_\theta : \mathcal{S} \rightarrow \mathcal{A}$ so as to maximize the objective function $\mathcal{J}(\theta) = \mathbb{E}_\tau \sum_t \gamma^t r_t$ where $\tau$ is a trajectory obtained from $\pi_\theta$ starting from state $s_0 \sim \rho_0$ and $r_t$ is the reward obtained along this trajectory at time $t$. We define the Q-value for policy $\pi$, $Q^\pi : \mathcal{S} \times \mathcal{A} \rightarrow \mathbb{R}$ as $Q^\pi(s, a) = \mathbb{E}_\tau \sum_t \gamma^t r_t$, where $\tau$ is a trajectory obtained from $\pi_\theta$ starting from $s$ and performing initial action $a$.

QD aims at evolving a set of solutions $\theta$ that are both diverse and high performing. To measure diversity, we first define a Behavior Descriptor (BD) space, which characterizes solutions in functional terms, in addition to their score $\mathcal{J}(\theta)$. We note $bd_\theta$ the BD of a solution $\theta$. The solution BD space is often designed using relevant features of the task. For instance, in robot navigation, a relevant BD is the final position of the robot. In robot locomotion, it may rather be the position and/or velocity of the robot center of gravity at specific times. From BDs, we define the diversity (or novelty) of a solution as measuring the difference between its BD and those of the solutions obtained so far.

Additionally, we define a *state Behavior Descriptor*, or state BD, noted $bd_t$. It is a set of relevant features extracted from a state. From state BDs, we define the BD of a solution $\theta$ as a function of all state BDs encountered by policy $\pi_\theta$ when interacting with the environment, as illustrated in Figure 1a. More formally, we note $bd_\theta = \mathbb{E}_\tau [f_{bd}(\{bd_1, \ldots, bd_T\})]$, where $T$ is the trajectory length and $f_{bd}$ is an aggregation function. For instance, $f_{bd}$ can average over state BDs or return only the last state BD of the trajectory. If we consider again robot navigation, a state BD $bd_t$ may represent the position of the robot at time $t$ and the solution BD $bd_\theta$ may be the final position of the robot. With state BDs, we measure the novelty of a state relatively to all other seen states.

The way we compute diversity at the solution and the state levels is explained in Section 4.

## 3 RELATED WORK

A distinguishing feature of our approach is that we combine diversity seeking at the level of trajectories using solution BDs $bd_\theta$ and diversity seeking in the state space using state BDs $bd_t$. The former is used to select agents from the archive in the QD part of the architecture, whereas the latter is used during policy gradient steps in the RL part, see Figure 1b. We organize the literature review below according to this split between two types of diversity seeking mechanisms.

Besides, some families of methods are related to our work in a lesser extent. This is the case of algorithms combining evolutionary approaches and deep RL such as CEM-RL (Pourchot & Sigaud, 2018), ERL (Khadka & Tumer, 2018) and CERL (Khadka et al., 2019), algorithms maintaining a population of RL agents for exploration without an explicit diversity criterion (Jaderberg et al., 2017) or algorithms explicitly looking for diversity but in the action space rather than in the state space like ARAC (Doan et al., 2019), P3S-TD3 (Jung et al., 2020) and DVD (Parker-Holder et al., 2020). We include CEM-RL as one of our baselines.

**Seeking for diversity and performance in the space of solutions** Simultaneously maximizing diversity and performance is the central goal of QD methods (Pugh et al., 2016; Cully & Demiris, 2017). Among the various possible combinations offered by the QD framework, Novelty Search with Local Competition (NSLC) (Lehman & Stanley, 2011b) and MAP-Elites (ME) (Mouret &

Clune, 2015) are the two most popular algorithms. In ME, the BD space is discretized into a grid of solution bins. Each bin is a niche of the BD space and the selection mechanism samples individuals uniformly from all bins. On the other hand, NSLC builds on the Novelty Search (NS) algorithm (Lehman & Stanley, 2011a) and maintains an unstructured archive of solutions selected for their local performance. Cully & Mouret (2013) augment the NSLC archive by replacing solutions when they outperform the already stored ones. In QD-RL, we build on the standard ME approach and on an augmented version of the NSLC algorithm where the population is selected from a global quality-diversity Pareto front inspired from Cully & Demiris (2017). Relying on these components, algorithms such as QD-ES and NSR-ES have been applied to challenging continuous control environments in Conti et al. (2018). But, as outlined in Colas et al. (2020), these approaches are not sample efficient and the diversity and environment reward functions could be mixed in a more efficient way. In that respect, the most closely related work w.r.t. ours is Colas et al. (2020). The ME-ES algorithm also optimizes both quality and diversity, using an archive, two ES populations and the MAP-ELITES approach. Using such distributional ES methods has been shown to be critically more efficient than population-based GA algorithms (Salimans et al., 2017), but our results show that they are still less sample efficient than off-policy deep RL methods as they do not leverage the policy gradient.

Finally, similarly to us, the authors of Shi et al. (2020) try to combine novelty search and deep RL by defining behavior descriptors and using an NS part on top of a deep RL part in their architecture. In contrast with our work, the transfer from novel behaviors to reward efficient behaviors is obtained through goal-conditioned policies, where the RL part uses goals corresponding to outcomes found by the most novel agents in the population. But the deep RL part does not contain a diversity seeking mechanism in the state space.

**Seeking for diversity and performance in the state space**   Seeking for diversity in the space of states or actions is generally framed into the RL framework. An exception is Stanton & Clune (2016) who define a notion of *intra-life novelty* that is similar to our *state novelty* described in Section 2. But their novelty relies on skills rather than states. Our work is also related to algorithms using RL mechanisms to search for diversity only, such as Eysenbach et al. (2018); Pong et al. (2019); Lee et al. (2019); Islam et al. (2019). These methods have proven useful in the sparse reward case, but they are inherently limited when the reward signal can orient exploration, as they ignore it. Other works sequentially combine diversity seeking and RL. The GEP-PG algorithm Colas et al. (2018) combines a diversity seeking component, namely *Goal Exploration Processes* (Forestier et al., 2017) and the DDPG deep RL algorithm (Lillicrap et al., 2015). This sequential combination of exploration-then-exploitation is also present in GO-EXPLORE (Ecoffet et al., 2019) and in PBCS (Matheron et al., 2020). Again, this approach is limited when the reward signal can help orienting the exploration process towards a satisfactory solution. These sequential approaches first look for diversity in the space of trajectories, then optimize performance in the state action space, whereas we do so simultaneously in the space of trajectories and in the state space.

Thus, as far as we know, QD-RL is the first algorithm optimizing both diversity and performance in the solution and in the state space, using a sample efficient off-policy deep RL method for the latter.

## 4   METHODS

QD-RL evolves a population of $N$ neural policies $(\theta_1, \ldots, \theta_N)$ so as to maximise their expected return as well as the diversity within the population. It relies on an archive storing all candidate solutions for selection and on a shared replay buffer storing all transitions $(s_t, a_t, r_t, bd_t)$ collected when training all agents. As illustrated in Figure 1b, one iteration consists of three successive phases: population selection among the solutions in the archive, mutation of the population, and evaluation of the newly obtained solutions. These phases are repeated until a maximum computational budget is reached or until convergence.

**Solution archive and selection mechanism.** In this study, we consider two different QD configurations for archive management and selection mechanism. While they rely on different assumptions, we show in Section 5 that QD-RL manages to solve the tasks efficiently in both cases.

In the PARETO setting, we maintain an unstructured archive of solutions $\theta$ found so far as well as their behavior descriptors $bd_\theta$. The archive is a limited size FIFO list. Furthermore, we implement insertion rules inspired from Cully & Mouret (2013) to prevent adding too close solutions in the BD

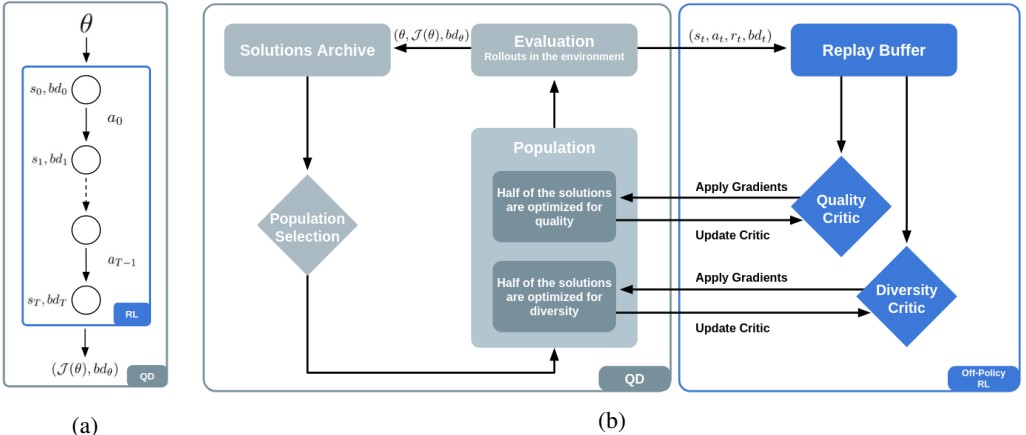

Figure 1: (a): The RL part of QD-RL operates over time steps while its QD part operates at the controller level, considering the MDP as a black box. (b) One QD-RL iteration consists of three phases: 1) A new population of solutions is sampled from the QD Pareto Front or from the Map-Elites grid. 2) These solutions are mutated by an off-policy RL agent: half of the solutions are optimized for quality and the other half for diversity. The RL agent leverages one shared critic for each objective. 3) The newly obtained solutions are evaluated in the environment. Transitions are stored in a replay buffer while final scores and BDs of solutions are stored in the QD archive.

space, unless they offer better performance (see Appendix B for more details). To select solutions from the archive, we compute a QD Pareto front of the solutions and sample the new population from this front. We use the $\mathcal{J}(\theta)$ objective function as the quality criterion, the diversity criterion is computed as the average Euclidean distance between $bd_\theta$ and its nearest neighbors in the archive.

In the ME setting, the behavior descriptor space is discretized into a Cartesian grid that constitutes the ME archive. We use the standard ME insertion criterion: when adding a solution $\theta$ into the grid, we compute its BDs and find the corresponding bin. If the bin is empty, we insert the solution. Otherwise, we add the solution only if its quality score is greater than the one of the contained solution. To select a new population, we sample solutions uniformly from the grid.

**Population mutation.** During one iteration, we mutate each solution in the population to optimize its quality with respect to the objective function, but also to optimize the population diversity. To decouple exploration from exploitation, we introduce two mutation operators optimizing respectively for quality and diversity. Both operators rely on an off-policy deep RL algorithm to compute gradients w.r.t. their respective criterion so as to to guide mutations towards improvement. While any off-policy algorithm supporting continuous action could be used, we rely on Twin Delayed Deep Deterministic Policy Gradient (TD3) (Fujimoto et al., 2018). To homogenize notations, we note $r_t^D$ for the novelty score and $r_t^Q$ for the environment reward. The novelty of a state $s_t$ at time step $t$ is computed as the mean Euclidean distance between its $bd_t$ and its nearest neighbors among BDs seen so far. More formally,

$$r_t^D = \frac{1}{k} \sum_{i=1}^{k} d\Big(bd_t, Neigh(A, bd_t, i)\Big),$$

where $d$ is the Euclidean distance, $k$ the number of nearest neighbors, and $Neigh(A, bd_t, i)$ the i-th nearest neighbor of state BD $bd_t$ in an archive $A$ of state BDs. This archive is filled with all states BDs collected by the population during training. As for the solutions archive, we use the insertion rules inspired from Cully & Mouret (2013). Technically, as this archive is used by the RL part of the algorithm, it is implemented as a sub-component of the experience replay buffer that receives the same state BDs but filters them with the insertion rules.

We introduce two critic networks $Q_w^Q$ and $Q_w^D$ that are shared over the population and used by TD3 to optimize solutions respectively for quality and diversity. Some reasons for sharing a critic among the population are given in Pourchot & Sigaud (2018). In our context, additional reasons come

from the fact that diversity is not stationary, as it depends on the current population. If each agent had its own diversity critic, since an agent may not be selected for a large number of generations before being selected again, its critic would evaluate it based on an outdated picture of the evolving diversity. We tried this solution and it failed. Besides, as diversity is relative to the population and not to an individual, it is more natural to have it centralized. A side benefit is that both shared critics become accurate faster as they combine the experience of all agents.

At each iteration, QD-RL mutates half the population to maximise quality and the other half to maximise diversity. As Colas et al. (2020), we use a 0.5 ratio to avoid biasing the algorithm towards exploration or exploitation. We leave for further study the automatic adjustment of this ratio, but we found that in practice QD-RL adequately balances both criteria during training.

To mutate a solution $\theta$, we sample a batch of transitions from the replay buffer and use stochastic gradient descent (SGD) where the gradient is computed as the TD3 policy gradient, see Equation (1). We respectively use the reward $r_t^Q$ when optimizing for quality, the reward $r_t^D$ when optimizing for diversity, and their corresponding critics. Note that while the $r_t^Q$ value is the one observed in the environment, the $r_t^D$ value changes across training time as new states are encountered. To deal with this non-stationary reward, we recompute a fresh score $r_t^D$ from the observed $bd_t$ every time a batch is sampled. The gradient computation over solutions is performed in parallel. Every time a policy gradient is computed, we also update the critics $Q_w^Q$ and $Q_w^D$. The critic gradients are computed in parallel and then averaged across the population. The global population update can be written as

$$
\begin{aligned}
\theta_i &\leftarrow \theta_i + \alpha \nabla_{\theta_i} \sum_{batch} Q_w^Q(s_t, \pi_{\theta_i}(s_t)), \ \ \forall i \leq N/2 \\
\theta_i &\leftarrow \theta_i + \alpha \nabla_{\theta_i} \sum_{batch} Q_w^D(s_t, \pi_{\theta_i}(s_t)), \ \ \forall i > N/2 \\
w &\leftarrow w - \frac{2\alpha}{N} \nabla_w \sum_{batch} \sum_{i=1}^{N/2} \left( Q_w^Q(s_t, a_t) - (r_t^Q + \gamma Q_{w'}^Q(s_{t+1}, \pi_{\theta_i}(s_{t+1}))) \right)^2 \\
w &\leftarrow w - \frac{2\alpha}{N} \nabla_w \sum_{batch} \sum_{i=N/2+1}^{N} \left( Q_w^D(s_t, a_t) - (r_t^D + \gamma Q_{w'}^D(s_{t+1}, \pi_{\theta_i}(s_{t+1}))) \right)^2,
\end{aligned}
\tag{1}
$$

where $w'$ are parameters of target critic networks. To keep notations simple, updates of the extra critic networks introduced in TD3, which serve to reduce the value estimation bias, do not appear in (1) although we use them in practice.

**Evaluation.** Once mutated, solutions are evaluated. To do so, each new solution $\theta$ performs trajectories with the resulting controller $\pi_\theta$ in the environment. All encountered transitions are stored in the replay buffer and we use the score $\mathcal{J}(\theta)$ and behavior descriptor $bd_\theta$ to place new solutions in the unstructured archive or in the MAP-Elites grid.

More details about the QD-RL algorithm and a pseudo-code are available in Appendices B and D.

## 5 EXPERIMENTS

In this section, we demonstrate the capability of QD-RL to solve challenging exploration problems. We implement it with TD3 and refer to this configuration as the QD-TD3 algorithm. Hyperparameters are described in Appendix B.5. We first analyse each component of QD-TD3 and demonstrate their usefulness on a simple example. Then, we show that QD-TD3 solves a more challenging control and exploration problem such as navigating the MUJOCO Ant in a large maze with a better sample complexity than its evolutionary competitors. Finally, we demonstrate that QD-TD3 can solve control problems when the behavior descriptor space is not aligned with the task.

### 5.1 POINT-MAZE AND POINT-MAZE-INERTIA: MOVE A POINT IN A MAZE

We first consider the POINT-MAZE environment in which a 2D material point agent must exit from the maze depicted in Figure 2a. In this environment, BDs are defined as the agent coordinates $(x_t, y_t)$ at time $t$ and an action corresponds to position increments along the $x$ and $y$ axis.

In order to exit the maze, the agent must find the right balance between exploitation and exploration. Although this environment may look simple due to its low dimensionality, it remains very challeng-

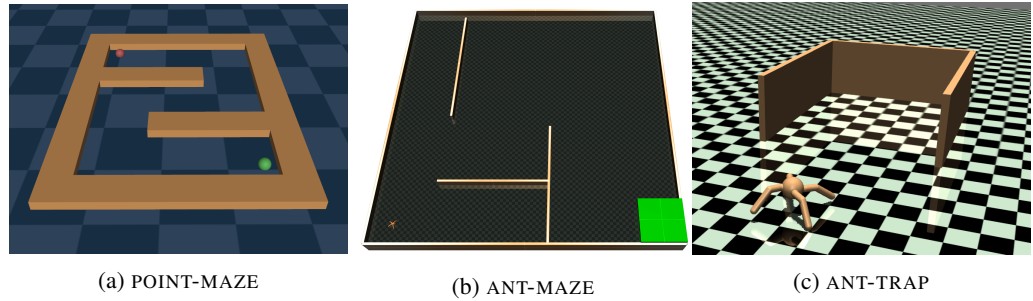

(a) POINT-MAZE      (b) ANT-MAZE      (c) ANT-TRAP

Figure 2: Evaluation environments. Though they may look similar, the state and action spaces in POINT-MAZE are two-dimensional, whereas they are $29 \times 8$ in ANT-MAZE.

ing for standard deep RL agents such as TD3, see Figure 4b. We also consider an identical task, POINT-MAZE-INERTIA, in which the point has momentum. In this case, BDs are unchanged but the state contains velocities in addition to positions, and the action is a force $(F_x, F_y)$. Despite the additional complexity, we observe equivalent performance.

We leverage these environments to perform an ablation study. First, we compare performance of the PARETO and the ME settings. Then, we measure performance when maximizing quality only. We call the resulting agent Q-TD3, but this is simply a multi-actor TD3. We also evaluate the agent performance when looking for solutions to maximize diversity only. We call the resulting agent D-TD3. Finally, we consider a D-TD3 + PARETO algorithm optimizing only for diversity but performing agent selection from the archive with a Pareto front. That is, it selects solutions for the next generation based on both their quality and diversity, but without optimizing the former. Table 1 summarises all the results and more figures are shown in Appendix C.3.

Table 1: Ablations and their performance in POINT-MAZE .

| Algorithm | Opt. Q | Opt. D | Selection mechanism | Episode return |
|---|---|---|---|---|
| QD-TD3 + ME | ✓ | ✓ | MAP-ELITES | $-\mathbf{24}\,(\pm 0)$ |
| QD-TD3 + PARETO | ✓ | ✓ | PARETO | $-27\,(\pm 1)$ |
| D-TD3 + PARETO | X | ✓ | PARETO | $-37\,(\pm 3)$ |
| D-TD3 | X | ✓ | None | $-111\,(\pm 53)$ |
| Q-TD3 + PARETO | ✓ | X | PARETO | $-128\,(\pm 0)$ |
| Q-TD3 | ✓ | X | None | $-130\,(\pm 2)$ |

## 5.2 ANT-MAZE: CONTROL AN ARTICULATED ANT TO SOLVE A MAZE

ANT-MAZE is modified from ANT-V2 of OpenAI Gym (Brockman et al., 2016) and is also used in Colas et al. (2020); Frans et al. (2018). In ANT-MAZE, a four-legged ant has to reach a goal zone located in the lower right part of the maze (colored in green in Figure 2b). Its initial position is sampled from a small circle around the initial point located in the extreme bottom left of the maze. Maze walls are organized so that following the gradient of distance to the goal drives the ant into a dead-end. As in POINT-MAZE, the reward is expressed as minus the Euclidean distance between the center of gravity of the ant and the center of the goal zone, thus leading to a strongly deceptive gradient. Note that in ANT-MAZE, the reward function does not contain the additional *control*, *healthy* and *contact* terms which are present in the reward function of ANT-V2 and help define an implicit curriculum that facilitates learning. The environment is considered solved when an agent obtains a score superior to $-10$, corresponding to reaching the goal zone. Since the agent must learn to control a body with 8 degrees of freedom in all directions to explore the maze and solve it, it is higher-dimensional than POINT-MAZE and more complex than the standard ANT-V2 where the ant should only go forward. The state contains the positions, angles, velocities and angular velocities of most ant articulations and center of gravity, and has 29 dimensions. An action specifies 8 continuous torque intensities applied to the 8 ant articulations. Episodes last 3000 time steps and the solution BD is the final position $(x_f, y_f)$ of the center of gravity of the ant, as in POINT-MAZE.

We compare QD-TD3 to five state-of-the-art evolutionary methods with a diversity seeking component: NSR-ES, NSRA-ES, NS-ES, ME-ES. While NS-ES and D-ME-ES optimize only for diversity, and Q-ME-ES optimizes only for quality, NSR-ES, NSRA-ES and QD-ME-ES optimize for both. To ensure fair comparison, we do not implement our own versions of these algorithms but reuse results from the ME-ES paper (Colas et al., 2020) and we make sure that our environment is rigorously the same. We also use as baselines SAC and TD3, two state-of-the-art RL methods for continuous control, and CEM-RL, an algorithm combining evolutionary methods and RL but without diversity seeking mechanisms. We chose CEM-RL as it shows state-of-the-art performance on standard MUJOCO benchmarks (Pourchot & Sigaud, 2018).

Table 2: Summary of compared algorithms on ANT-MAZE. The **Final Perf.** is the minimum distance to the goal at the end of an episode over the population.

| **Algorithm** | **Final Perf.** ($\pm$ std) | **Steps to -10** | **Ratio to** QD-TD3 |
|---|---|---|---|
| QD-TD3 PARETO | $-4\,(\pm3)$ | 1.15e8 | 1 |
| QD-TD3 ME | $-7\,(\pm7)$ | 1.15e8 | 1 |
| QD-TD3-SUM | $-5\,(\pm3)$ | 2.5e8 | 2 |
| D-TD3 | $-2\,(\pm0)$ | 3.5e8 | 3 |
| Q-TD3 | $-26\,(\pm0)$ | $\infty$ | $\infty$ |
| SAC | $-59\,(\pm1)$ | $\infty$ | $\infty$ |
| CEM-RL | $-26\,(\pm0)$ | $\infty$ | $\infty$ |
| NS-ES | $-4\,(\pm0)$ | 2.4e10 | 209 |
| NSR-ES | $-26\,(\pm0)$ | $\infty$ | $\infty$ |
| NSRA-ES | $-2\,(\pm1)$ | 2.1e10 | 182 |
| QD-ME-ES | $-5\,(\pm1)$ | 2.4e10 | 209 |
| D-ME-ES | $-5\,(\pm0)$ | 1.9e10 | 165 |
| Q-ME-ES | $-26\,(\pm0)$ | $\infty$ | $\infty$ |

### 5.3 ANT-TRAP: CONTROL AN ARTICULATED ANT TO AVOID A TRAP

ANT-TRAP also derives from ANT-V2 and is inspired from Colas et al. (2020). In ANT-TRAP, the four-legged ant initially appears in front of a trap and must avoid it in order to run as fast as possible in the forward direction along the $x$-axis. The trap consists of three walls that form a dead-end directly in front of the ant, leading to a strongly deceptive gradient. As in previous environments, the behavior space is defined as the final position $(x_f, y_f)$ of the center of gravity of the ant. The peculiarity of this environment compared to the others is that here, the reward is expressed as the ant's forward velocity and is thus not fully aligned with the BDs. In contrast to ANT-MAZE, here the reward function is the same as the one used in ANT-V2: it contain the additional features that facilitate training. We can thus test the coverage capacity of our method in the case of a partial alignment. The action and state spaces are the same as in ANT-MAZE and the episodes last 1000 time steps. In this benchmark, we compare QD-TD3 to SAC, TD3, CEM-RL and QD-TD3-SUM. We could not compare it to the ME-ES family as these algorithms require 1000 CPUs.

## 6 DISCUSSION

**Ablation study.** The ablation study in Table 1 shows that: 1) When maximising quality only, Q-TD3 with or without a selection mechanism fails due to the deceptive nature of the reward. Figure 4 also shows the poor BD space coverage resulting from the lack of a diversity component. 2) When maximising diversity only, D-TD3 fails without selection mechanism but adding PARETO or ME selection is enough to exit the maze. 3) When optimizing quality and diversity, the agent finds a more optimal trajectory to the maze exit, resulting in higher returns. 4) While ME has slightly less variance, both QD methods solve the problem with high returns. This study validates the usefulness of the QD-RL components: 1) optimizing for diversity is required to overcome the deceptive nature of the reward; 2) a proper population selection mechanism is required to ensure proper convergence, and adding quality optimization provides better asymptotic performance.

**Comparison to neuroevolution with a diversity seeking component competitors in Ant-Maze.**
Table 2 compares QD in terms of sample efficiency to Deep Neuroevolution algorithms with a

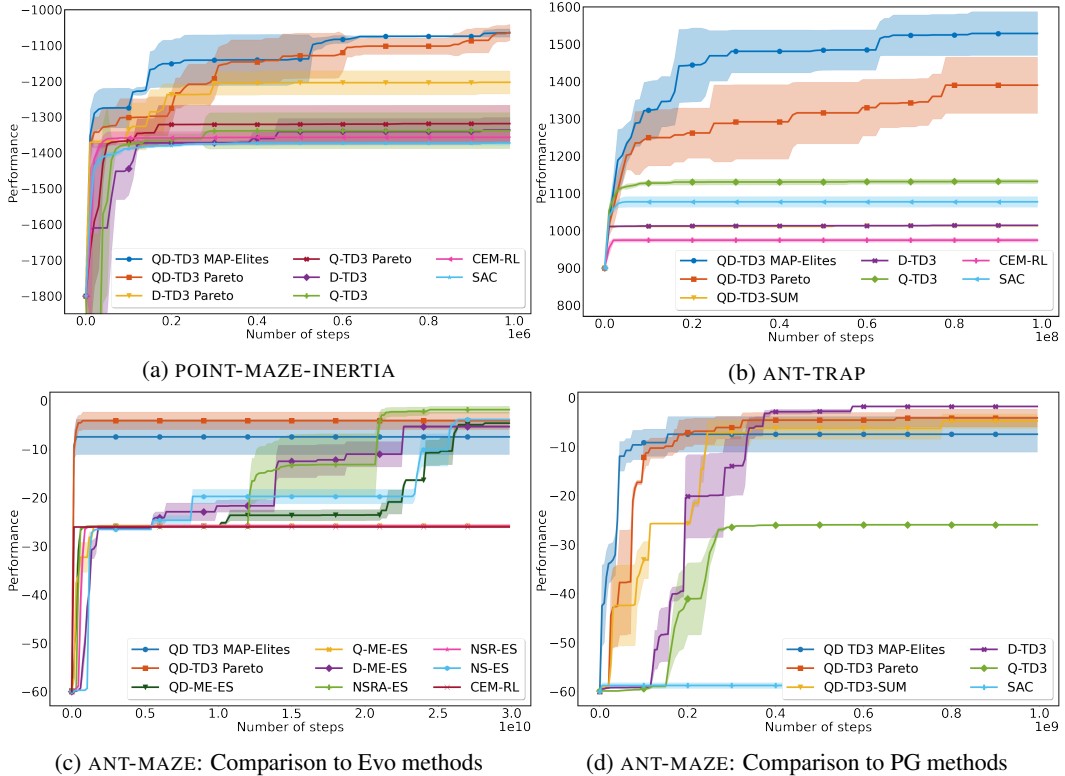

Figure 3: Learning curves of QD-TD3 versus ablations and baselines. In POINT-MAZE and ANT-TRAP, the performance is the highest return. In ANT-MAZE, it is minus the highest distance to the goal. In 3a and 3b, all results are averaged over 5 seeds. In 3c and 3d, curves show the best 2 seeds out of 5 for all algorithms.

diversity seeking component in ANT-MAZE. QD-RL is run on 10 CPU cores during 2 days while its competitors used several hundreds of CPU cores for the same duration. Nonetheless, QD-RL matches the asymptotic performance of ME-ES using two orders of magnitude less samples, thus explaining the lower resource requirements. We see three reasons for the improved sample efficiency of QD-RL: 1) QD-RL leverages a replay buffer and can re-use each sample several times. 2) QD-RL leverages a notion of novelty defined at the state level and thus can exploit all collected transitions to maximize both quality and diversity. For example, in ANT-MAZE, a trajectory brings 3000 samples to QD-RL while standard QD methods would consider it as a unique sample. 3) RL exploits the analytical gradient between the neural network weights and the resulting policy action distribution and thus estimates only the impact of the distribution on the return, whereas standard QD methods estimate directly the impact of the weights onto the return.

**Performance assessment when BDs are not fully aligned with the task in Ant-Trap.** Finally, we evaluated QD-RL in the ANT-TRAP environment. The difference from previous environments is that reward is computed as the ant forward velocity at time $t$ and not as an Euclidean distance to a target. As the BD remains the ant center of gravity, it is not directly aligned with the task. Nonetheless, we observe in Figure 4c that optimizing only for quality results in sub-optimal performance while QD-RL finds the optimal strategy by avoiding the trap and running beyond.

**Mixing quality and diversity updates.** To validate our design choices, we also introduce a variant of QD-TD3 called QD-TD3-SUM, where a unique critic $Q_w$ is updated considering at every time step the sum $r_t$ of the quality reward and the diversity reward: $r_t = r_t^Q + r_t^D$. We tested QD-TD3-SUM on ANT-MAZE and ANT-TRAP. QD-TD3-SUM completely fails to learn anything in ANT-TRAP. It manages to solve the task in ANT-MAZE but requires more samples than QD-TD3. In fact, the quality and diversity rewards may give rise to gradients pointing to opposite directions. For instance, at the beginning of training in ANT-TRAP, the quality reward drives the ant forward whereas the diversity

reward drives it backward so as to escape the trap and explore the environment. Therefore, both rewards may often anneal each other, preventing any learning. This situation, while rarer, also happens in the ANT-MAZE environment and reduces sample efficiency.

**Comparison to sample efficient methods without a diversity seeking component.** We compared QD-TD3 to TD3 and SAC and CEM-RL in both ANT-MAZE and ANT-TRAP benchmarks. As expected, TD3 converges quickly to the local minimum of performance resulting from being attracted in the dead-end by the deceptive gradient in both environments. While we may expect SAC to demonstrate better exploration properties as it regularizes entropy, it also converges to that same local minimum in ANT-TRAP. Surprisingly, SAC with the standard hyper-parameters used for MuJoCo benchmarks does not learn in ANT-MAZE. It may come from the lack of the additional features in the reward function. CEM-RL combines RL for sample efficiency and cross-entropy methods for a better exploration in the parameter space. Indeed, CEM-RL was shown to outperform RL algorithms in standard MuJoCo environments. Despite this additional exploration mechanism, CEM-RL also quickly converges to the dead-end in both benchmarks, thus confirming the need for a dedicated diversity seeking component.

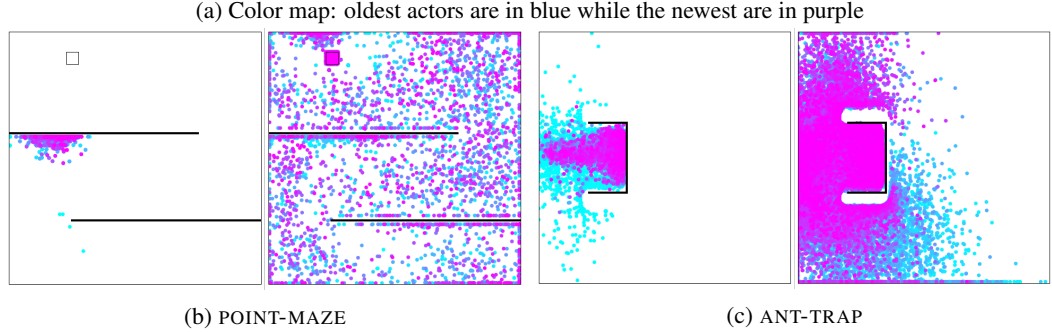

(a) Color map: oldest actors are in blue while the newest are in purple

(b) POINT-MAZE          (c) ANT-TRAP

Figure 4: Coverage maps of Q-TD3 (left) and QD-TD3 (right) in POINT-MAZE and ANT-TRAP.

## 7    CONCLUSION

In this paper, we proposed a novel way to deal with the exploration-exploitation trade-off by combining a quality-seeking component, a diversity-seeking component and a selection component inspired from the Quality-Diversity literature. Crucially, we showed that quality and diversity could be optimized with off-policy reinforcement learning algorithms, resulting in a significantly better sample efficiency. We showed experimentally the effectiveness of the resulting QD-RL framework, which can solve, in two days using 10 CPUs, problems that were previously out of reach without a much larger infrastructure. Key components of QD-RL are selection through a Pareto front or a Map-Elites grid and the search for diversity in a behavior descriptor space. Admittedly, this behavior descriptor space is hand designed. There are attempts to automatically obtain it through unsupervised learning methods (Péré et al., 2018; Paolo et al., 2019), but defining this space is often a trivial design choice that can alleviate the need to carefully shape reward functions. In the future, we intend to extend this approach to problems where the reward function can be decomposed into several loosely dependent components, such as standing, moving forward and manipulating objects for a humanoid agent, or to multi-agent reinforcement learning problems. In such environments, we could replace the maximization of the sum of reward contributions with a multi-criteria selection from a Pareto front, where diversity would be only one of the considered criteria. Additionally, we would like to make profit of the diversity of the solutions we obtain to build repertoires of behaviors, as it is one of the main features of the Quality-Diversity methods.

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

APPENDICES

## A  THE TD3 AGENT

The Twin Delayed Deep Deterministic (TD3) agent Fujimoto et al. (2018) builds upon the Deep Deterministic Policy Gradient (DDPG) agent Lillicrap et al. (2015). It trains a deterministic actor $\pi_\phi : \mathcal{S} \to \mathcal{A}$ directly mapping observations to continuous actions and a critic $Q_\theta : \mathcal{S} \times \mathcal{A} \to \mathbb{R}$ taking a state $s$ and an action $a$ and estimating the average return from selecting action $a$ in state $s$ and then following policy $\pi_\phi$. As DDPG, TD3 alternates policy evaluation and policy improvement operations so as to maximise the average discounted return. In DDPG, the critic is updated to minimize a temporal difference error during the policy evaluation step which induces an overestimation bias. TD3 corrects for this bias by introducing two critics $Q_{\theta_1}$ and $Q_{\theta_2}$. TD3 plays one step in the environment using its deterministic policy and then stores the observed transition $(s_t, a_t, r_t, s_{t+1})$ into a replay buffer $\mathcal{B}$. Then, it samples a batch of transitions from $\mathcal{B}$ and updates the critic networks. Half the time it also samples another batch of transitions to update the actor network.

Both critics are updated so as to minimize a loss function which is expressed as a mean squared error between their predictions and a target:

$$L^{critic}(\theta_1, \theta_2) = \sum_{\text{batch}} (Q_{\theta_1}(s_t, a_t) - y_t)^2 + (Q_{\theta_2}(s_t, a_t) - y_t)^2, \tag{2}$$

where the common target $y_t$ is computed as:

$$y_t = r_t + \gamma \min_{i=1,2} Q_{\theta_i}(s_{t+1}, \pi_\phi(s_{t+1}) + \epsilon), \;\; \epsilon \sim \mathcal{N}(0, I). \tag{3}$$

The Q-value estimation used to compute target $y_t$ is taken as minimum between both critic predictions thus reducing the overestimation bias. TD3 also adds a small perturbation $\epsilon$ to the action $\pi_\phi(s_{t+1})$ so as to smooth the value estimate by bootstrapping similar state-action value estimates.

Every two critics updates, the actor $\pi_\phi$ is updated using the deterministic policy gradient also used in DDPG Silver et al. (2014). For a state $s$, DDPG updates the actor so as to maximise the critic estimation for this state $s$ and the action $a = \pi_\phi(s)$ selected by the actor. As there are two critics in TD3, the authors suggest to take the first critic as an arbitrary choice. Thus, the actor is updated by minimizing the following loss function:

$$L^{actor}(\phi) = -\sum_{\text{batch}} Q_{\theta_1}(s_t, \pi_\phi(s_t)). \tag{4}$$

Policy evaluation and policy improvement steps are repeated until convergence. TD3 demonstrates state of the art performance on several MUJOCO benchmarks. In this study, we use it to update the population of actors for both quality and diversity.

## B  QD-RL ADDITIONAL PRACTICAL DETAILS

### B.1  COMPUTATIONAL DETAILS

We consider populations of $N = 4$ actors for both POINT-MAZE and POINT-MAZE-INERTIA environments and $N = 10$ actors for ANT-MAZE and ANT-TRAP. We use 1 CPU thread per actor and parallelization is implemented with the Message Passing Interface (MPI) library. Our experiments were run on a standard computer with 10 CPU cores and 100 GB of RAM although the maximum RAM consumption per experiment at any time never exceeds 10GB due to an efficient and centralized management of the archive that stores all solutions. An experiment on POINT-MAZE or POINT-MAZE-INERTIA typically takes between 2 and 3 hours while an experiment on ANT-MAZE or ANT-TRAP takes about 2 days. Note that these durations can vary significantly depending on the type of CPU used. We did not use any GPUs.

### B.2   QD Implementation details

We consider two settings for the Quality Diversity part of our algorithm.

In the PARETO setting, we rely on a FIFO archive of solutions. We chose a maximum archive size of 10.000 for all environments. When a new solution $\theta$ is obtained after the mutation phase, we compute the mean distance between its BD $bd_\theta$ and the BDs of its K nearest neighbors in the archive. We used $K = 10$ for all environments. We also used an Euclidean distance for all environments. We add the new solution in the archive if the computed distance is superior to a threshold of acceptance. See Table 3 for the different values of this threshold depending on the environment. During the selection phase, we compute a QD Pareto front of all solutions contained in the archive and sample the new population from the front. If the Pareto front contains less than $N$ actors, we select them all, remove them, compute the Pareto front over the remaining actors and sample again from it, and so on until we get $N$ actors.

In the ME setting, we use a Map Elites (ME) grid as archive of solutions. We assume that the BD space is bounded and can be discretized in an Cartesian grid. We discretize each dimension into $m$ meshes, see Table 3 for the value of $m$ depending on the environment. Thus the number of bins in the ME grid equals $m$ times the number of dimensions of the BDs. When a new solution $\theta$ is obtained after the mutation phase, we seek for the bin that corresponds its BD $bd_\theta$. If the bin is empty, we add the solution in it, otherwise we replace the solution already contained by the new one if its score $\mathcal{J}(\theta)$ is greater than the one of the already contained solution. During selection, we sample solutions uniformly from the ME grid. The ME archive management and selection mechanism tends to store less solutions than the PARETO method and does not need to compute a Pareto Front which makes QD-RL ME less computation and memory costly than QD-RL PARETO.

### B.3   Diversity reward computation

The RL part of QD-RL optimizes solutions for quality but also for diversity at the state level. As the QD part stores solutions and performs population selection based on solution BDs $bd_\theta$, the RL part optimizes the solutions so as to encourage them to visit states with novel state BDs. The novelty of a state BD $bd_t$ is expressed through a diversity reward $r_t^D$. In practice, we maintain in parallel of the replay buffer a FIFO storage of the states BDs encountered so far. In practice, the maximum size of this storage is never reached. However its maximum theoretical size is chosen to be the same as the solutions archive. When a transition $(s_t, a_t, bd_t, r_t, s_{t+1})$ is sampled and stored in the Replay Buffer, we also add $bd_t$ in this storage. For the archive of solutions presented in the PARETO setting, we add a state BD in this storage only if its mean Euclidean distance to its $K$ nearest neighbors in the storage is greater than an acceptance threshold. The values of $K$ and of the threshold are given in Table 3. This mechanism ensures that we do not keep too close state BDs and thus controls this storage size. When a batch of transitions $(s_t, a_t, bd_t, r_t, s_{t+1})$ is collected during the mutation phase, we recompute fresh diversity rewards $r_t^D$ as the mean Euclidean distance between the sampled states BDs $bd_t$ and their $K$ nearest neighbors in the states BDs storage. The value of $K$ is the same as the one used for the insertion rule.

### B.4   Computational cost and parallel implementation

The main costs of the algorithm come from backpropagation during the update of each agent, and to the interaction between agents and the environment. These costs scale linearly with the population size but, as many other population-based methods, the structure of QD-RL lends itself very well to parallelization. We leverage this property and parallelize our implementation so as to assign one agent per CPU core. Memory consumption also scales linearly with the number of agents. To reduce this consumption, we centralize the solution archive on a master worker and distribute data among workers when needed. With these implementation choices, QD-RL only needs a very accessible computational budget for all experiments.

## B.5 HYPER-PARAMETERS

Table 3 summarizes all hyper-parameters used in experiments (see Figure 3). Most of these hyper-parameters values are the original ones from the TD3 algorithm.

Table 3: QD-TD3 Hyper-parameters

| Parameter | Point Maze | Ant Maze | Ant Trap |
|---|---|---|---|
| **TD3** | | | |
| Optimizer | Adam | Adam | Adam |
| Learning rate | $6.10^{-3}$ | $3.10^{-4}$ | $3.10^{-4}$ |
| Discount factor $\gamma$ | 0.99 | 0.99 | 0.99 |
| Replay buffer size | $10^6$ | $5.10^5$ | $5.10^5$ |
| Hidden layers size | 64/32 | 256/256 | 256/256 |
| Activations | ReLU | ReLU | ReLU |
| Minibatch size | 256 | 256 | 256 |
| Target smoothing coefficient | 0.005 | 0.005 | 0.005 |
| Delay policy update | 2 | 2 | 2 |
| Target update interval | 1 | 1 | 1 |
| Gradient steps ratio | 4 | 0.1 | 0.1 |
| **Quality-Diversity** | | | |
| Archive size | 10000 | 10000 | 10000 |
| Threshold of acceptance | 0.0001 | 0.1 | 0.1 |
| K-nearest neighbors | 10 | 10 | 10 |
| **MAP-Elites** | | | |
| Number of bins per dimension | 5 | 7 | 6 |

## C ENVIRONMENTS ANALYSIS

In this section, we propose a more thorough analysis of the proposed environments. We highlight why these environments are hard to solve for classical deep RL agents without a diversity mechanism and show the impact of the different components of our algorithm.

### C.1 ENVIRONMENTS DETAILS

In POINT-MAZE, the observation and state BD are the agent coordinates $(x_t, y_t)$ at time $t$. The actions features $(\delta x, \delta y)$ are position increments along the $x$ and $y$ axes. The solution BD is the final position $(x_f, y_f)$ of the agent, as in Conti et al. (2018). The initial position of the agent is sampled from a small zone at the bottom of the maze. The exit area is a square located at the top left of the maze, the episode ends once this exit square is reached or after 200 time steps. The reward is computed as minus the Euclidean distance between the agent position and the maze exit. This reward leads to a deceptive gradient signal: following it would lead the agent to stay stuck by the second wall in the maze, as shown in Figure 5 of Appendix C. In POINT-MAZE-INERTIA, the BD space remains the same however the point velocity $(v_x, v_y)$ is appended to the state to preserve the Markov property and the action corresponds to forces $(F_x, F_y)$.

In POINT-MAZE and POINT-MAZE-INERTIA, the dimensions of the observation spaces are equal to 2 and 4 respectively, and the dimensions of the action spaces are both equal to 2. Both environments are very similar with low-dimensional state and action spaces, and only differ by the fact that POINT-MAZE-INERTIA moves a ball subject to inertia. By contrast, in ANT-MAZE and ANT-TRAP, the dimensions of the observation spaces are respectively 29 and 113 while the action spaces dimensions are both equal to 8, making these two environments much more challenging as they require larger controllers.

The ANT-TRAP environment also differs from mazes as it is open-ended, i.e., the space to be explored by the agent is unlimited, in contrast to other environments where this space is restricted by the maze walls. In this case, the state BD corresponds to the ant position that is clipped to remain in a given range. On the $y$-axis, this range is defined so as to leave a space corresponding to the width of the trap on both trap sides. On the $x$-axis, this range begins slightly behind the starting position of the ant and is large enough to let it accelerate along this axis. Figure 4 proposes a visual representation of the BD space in ANT-TRAP.

## C.2   DECEPTIVE GRADIENTS

Figure 5 highlights the deceptive nature of the POINT-MAZE and the ANT-MAZE objective functions by depicting gradient fields in both environments. Similarly, POINT-MAZE-INERTIA and ANT-TRAP also present strongly deceptive gradients.

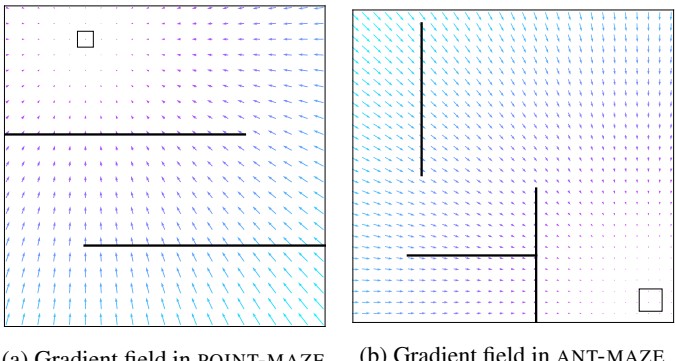

(a) Gradient field in POINT-MAZE    (b) Gradient field in ANT-MAZE

Figure 5: Gradients maps on POINT-MAZE and ANT-MAZE. The black lines represent the maze walls, the arrows depict gradient fields and the square indicates the maze exit. Both settings present highly deceptive gradients: naively following them leads into a wall.

## C.3   EXPLORATION IN POINT-MAZE FOR ALL ABLATIONS

Figure 6 summarizes the coverage maps of the POINT-MAZE environment by the different algorithms presented in the ablation study (see Table 1). A dot in the figure corresponds to the final position of an agent after an episode. The color spectrum highlights the course of training: agents evaluated early in training are in blue while newer ones are represented in purple. Figure 6 corresponds to the map coverage for one seed, we chose a representative seed.

QD-TD3 (PARETO andME) and D-TD3+PARETO almost cover the whole BD space including the objective. D-TD3 also finds the objective but is not constant through all seeds and does not uniformly cover the BD space. Unsurprisingly, Q-TD3+PARETO and Q-TD3 present very poor coverage maps, both algorithms optimize only for quality and the Pareto selection mechanism does not contribute anything in this setting. Interestingly, all algorithms optimizing for diversity find the maze exit. However, as shown in Table 1, algorithms that also optimize for quality (QD-TD3) or that present structural pressure toward quality (D-TD3+PARETO) are able to refine their trajectory through the maze and obtain significantly better performance.

Figure 7 summarizes the coverage maps of the ANT-TRAP environment by QD-TD3+ME, QD-TD3+PARETO and TD3. Interestingly, the pattern of QD-TD3+ME differs from that of QD-TD3+PARETO. We hypothesize that this behavior reflects the grid structure of the ME archive.

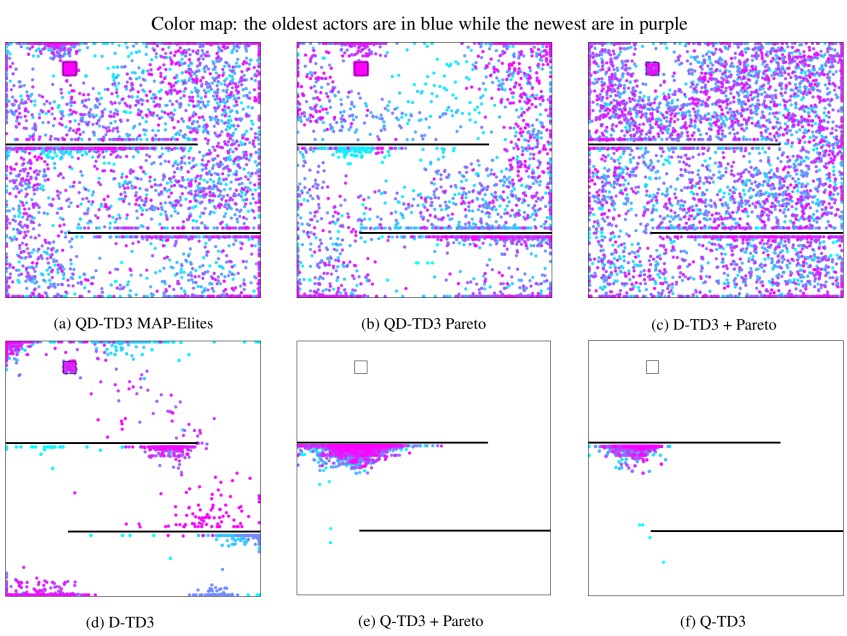

Figure 6: Coverage map of the POINT-MAZE environment for all ablations. Each dot corresponds to the position of an agent at the end of an episode. Dots corresponding to the oldest actors are in blue while the newest are in purple.

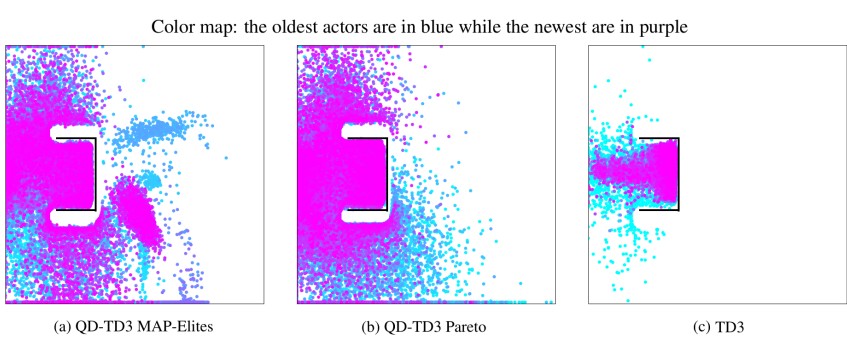

Figure 7: Coverage map of the ANT-TRAP environment. Each dot corresponds to the position of an agent at the end of an episode. Dots corresponding to the oldest agents are in blue while the newest are in purple.

# D QD-RL PSEUDO CODE

---

**Algorithm 1:** QD-RL

---

**Given:** N, max_steps, gradient_steps_ratio,
**Initialize:** Archive $A$, Replay Buffer $\mathcal{B}$, $N$ actors $\{\pi_{\theta_i}\}_{i=\{1,\dots,N\}}$, 2 critics $Q_{\theta^D}$ and $Q_{\theta^Q}$, BD Storage $S$

$total\_steps, actor\_steps = 0, 0$ // Step counters

// Parallel evaluation of the initial population
**for** $j \leftarrow 1$ **to** $N$ **do**
    Play one episode with actor $\pi_{\theta_j}$ and store all transitions in $\mathcal{B}$
    Get episode length $T$, discounted return $R$ and state BDs $\{bd_1, \dots, bd_T\}$
    Store BDs $\{bd_1, \dots, bd_T\}$ in $S$
    Compute $bd_\theta = f_{bd}(\{bd_1, \dots, bd_T\})$ and add the tuple $(R, bd_\theta, \theta_j)$ in the archive $A$
    $actor\_steps \leftarrow actor\_steps + T$
**end**

// Algorithm main loop
**while** $total\_steps < max\_steps$ **do**
    // Select new generation
    Get $N$ actors $\pi_{\theta_i}, i \in \{1, \dots, N\}$ from the Pareto front or from the MAP-Elites grid
    $gradient\_steps = int(actor\_steps \times gradient\_steps\_ratio)$
    $actor\_steps = 0$

    // Perform in parallel population update and evaluation
    **for** $j \leftarrow 1$ **to** $N$ **do**
        // Update the population
        **for** $i \leftarrow 1$ **to** $gradient\_steps$ **do**
            Sample batch of $(s_t, a_t, r_t, s_{t+1}, bd_t)$ from $\mathcal{B}$

            // First half is updated to maximise diversity
            **if** $j \leq N//2$ **then**
                Compute novelty reward as $r_t^D$ from $bd_t$ and $S$
                Update $\pi_{\theta_j}$ for diversity
                Compute novelty critic gradient locally
                Average between threads novelty critic gradients
                Update novelty critic $Q_w^D$
            **end**

            // Second half is updated to maximise quality
            **else**
                Update $\pi_{\theta_j}$ for quality
                Compute quality critic gradient locally
                Average between threads quality critic gradients
                Update quality critic $Q_w^Q$
            **end**
        **end**

        // Evaluate the updated actors
        Play one episode with actor $\pi_{\theta_j}$ and store all transitions in $\mathcal{B}$
        Get episode length $T$, discounted return $R$ and state BDs $\{bd_1, \dots, bd_T\}$
        Store BDs $\{bd_1, \dots, bd_T\}$ in $S$
        Compute $bd_\theta = f_{bd}(\{bd_1, \dots, bd_T\})$ and add the tuple $(R, bd_\theta, \theta_j)$ in the archive $A$
        $actor\_steps \leftarrow actor\_steps + T$
    **end**

    $total\_steps \leftarrow total\_steps + actor\_steps$ // Update total time steps
**end**

---