# OpenReview forum: "Sample efficient Quality Diversity for neural continuous control"
_ICLR.cc/2021/Conference — Reject_

### Official Review · AnonReviewer4 · 2020-10-21
**A promising idea that is lacking sufficient experimental evidence**

**Rating:** 6
**Confidence:** 3

**Review:**

Summary
-------
The paper proposes a way to use off-policy reinforcement learning (RL) in the population mutation step of the quality-diversity (QD) framework. To the best of my knowledge (which is not too extensive in the realm of QD literature) the idea to use two critics, one for quality, one for diversity and use them to mutate the populations of agents these two directions, is novel and looks interesting.


Strengths
---------
* The general flow of the paper is clear and is easy to follow.
* Related work gives a good overview of diversity-seeking approaches.
* The idea to use off-policy RL for mutations in two different directions is interesting and intuitively should indeed result in sample complexity reduction as stated in the title of the paper.


Weaknesses and questions
------------------------
1) As someone relatively new to QD I would benefit from references (somewhere early on, in the introduction) to core research of these methods. Each of the first three sentences in the second paragraph of the introduction begs for references to support the statements that are being made: "The QD literature generally builds on black-box optimization approaches such as evolutionary algorithms to optimize a population of solutions [REF]. These algorithms often rely on random mutations and/or crossover to explore small search spaces but struggle when confronted to higher-dimensional problems [REF]. As a result, QD methods often scale poorly to large sequential decision problems with continuous state and action spaces [REF]".

2) Why diversity is measured in terms of BD and not directly in the parameter space or state space? What makes it preferable? If this is something that generally done in QD literature an explanation why this is the way would do good. If this is a novelty introduced in this paper, it warrants a full and detailed justification.

3) I would like it to be more clearly stated what was the prior work on combining QD with RL, or whether your work is first such attempt. While QD was attempted with other methods, this paper claims "QD-RL is the first algorithm optimizing both diversity and performance in the solution and in the state space, using a sample efficient deep RL method for the latter". The use of RL is alluded to in the second paragraph of section 3, but, taking into account that combination of RL and QD is the core claim of this paper this combination in other works deserves better description and not just a mention. For example was the core idea of using off-policy RL with two critics used in some prior work? Maybe applied to another space or part of the optimization process? If so, this should be mentioned.

4) How is diversity calculated? The "problem statement" section explains that "The measure of diversity relies on the definition of a DB", but is not clearly defined.

5) [ CRITICAL ] The core claim of the paper (that QD-RL is more sample efficient) comes from the results presented on the Figure 2b and it said that the experimental environment is an exact copy of the environment used in Colas et al. (2020) and the results reported in that paper are taken as the point of comparison. I would like to clarify a couple of points stemming from this.

5.1) Why the "Deceptive Humanoid" was not evaluated in this paper? Would the sample complexity reduction be seen in this environment as well if tested? (I can see that "Deceptive Humanoid" was replaced by "Ant Trap" but since Ant Trap was not evaluated in the referenced paper it does not allow for comparison)

5.2) On Figure 6a in Colas et al. (2020) [https://arxiv.org/pdf/2003.01825.pdf] the reported performance of NSRA-ES is -10, while in this work it is shown to be -2. How did this difference occur? Same applied to ME-ES Explore, which is -12 in the original paper and reported as -6 in this work. Similar discrepancy seems to affect other method as well.

5.3) In the original paper sample complexity is shown as the number of generations, while in this work it is shows in numbers of steps. Is there a fixed mapping between the two and is it same for all algorithms?


6) [ CRITICAL ] While there seems to be an abundance of other QD-based methods that were tried on similar environments, these experimental results, most importantly their sample complexities, are not reported in this work and not compared with the proposed method.

7) In the ablation study summarized in Table 1, the candidate algorithm (QD-TD3) is the only one that was tested with MAP-Elites selection mechanism. What would be the performance of the competitor methods ("D-TD3" and "Q-TD3") if MAP-Elites would be used with them as selection mechanism?


Recommendation and justification
--------------------------------
I like the idea presented in this paper and my intuition regarding the benefit or using off-policy RL is aligned with the authors' in that it should lead to higher sample efficiency due to the use of a replay buffer and direct access of the policy gradient. However the experimental results compare the performance with only one other study and with only one task within that study (using "Ant Maze", but skipping "Deceptive Humanoid"). This begs the question how the sample complexity of QD-RL compares to other methods that apply QD to similar tasks and environments.

Without such a wider comparison that would confirm the main claim of the paper it is hard for me to recommend this paper for acceptance as the experimental evidence for the main claim is not sufficient in my opinion.


Additional remarks
------------------
It was not clear to me the role of the "Ratio to QD-TD3" column in Table 2. Maybe a brief explanation of its significance in the table caption would help.


-------------------- UPDATE Nov 30 --------------------
---------------------------------------------------------------

I find the additional experimental work that was carried out for the revised version of the manuscript to be a step in the right direction which has provided more confidence in the claim of the paper.

The presentation of results is a bit hard to follow... it's a bit hard to put finger on what is exactly the issue. One thing I would suggest is making the names of the methods a bit more telling, deeper into the paper it becomes hard to track which abbreviation stands for which method. It also messy when some methods are reffed to by a particular RL algorithm name "QD-TD3" some by just mentioning RL "CEM-RL" some just as "SAC" and this naming convention breaks when evolutionary methods are mentioned. Maybe for someone who works with these methods a lot it is easy to keep track, but not for a reader not directly involved with the QD field.

Why in Table 2 the third column was changed from "Step to -5" to "Steps to -10"? How this number was picked?

Figure 3, which is the main evidence for the main claim of the paper only appear in Discussion, leaving an impression that this figure is not that important and is a bit of a side-note.

I sympathize with the lack of computational resources, which makes it very hard to compete, but if a 1-to-1 comparison with competitors if at all feasible, it would be worth it. If your methods is as strong as it seems it is, then it will beat the competition across the board in term of sample complexity and send a clean and powerful message that utilizing RL with QD is the way to go.

At this point it might be that the main issue with this work not receiving higher scores lies not in the idea or experimental work, but in presentation. Try taking a couple of you colleagues who are not familiar with their work and observe how they read it, notice the moment when they start loosing focus. Your text jumps from one message to another making the overall narrative not as streamlined as it could. This might be reason for reviews like R3's, where, it seems, the reader gets lost and comes out without clear understanding of the outcomes of your experiments and how these support your claims.

To summarize I still find the idea clever and with the new evidence I am more confident that the claim of the paper might hold in general. Since experimental evidence was my main concern and now there is more of it, I am upping my score from 4 to 6 - "Marginally above acceptance threshold".

---

> ### Author Response · Authors · 2020-11-24
> **Response to Reviewer4 (2/2)**
>
> The reviewer also asks whether there is a fixed mapping between generations and steps, and if it is the same for all algorithms. This mapping is fixed and as follows: The number of time steps during an entire run is computed as the number of generations, times the population size, times the number of episodes played by each agent during one generation, times the number of time steps per episode. In this paper we wanted to compare all methods on the basis of the number of interactions with the environment (i.e. the number of time steps), which is generally considered to be a cost that one wants to minimize. Actually, by talking with the authors of the Colas et al. 2020 paper to investigate the generation to steps ratio to answer the question, we spotted a mistake in the previous version. We thought the ratio was 10 billion steps per generation, it is in fact 30 billion steps per generation. Thus, QD-RL is more sample efficient wrt ME-ES than we thought.
>
> The reviewer seems to be asking us to report the sample complexities of all methods that were tried on similar environments. Can the reviewer be more specific about this query? We don’t know up to which extent we should consider other environments as similar to ours, nor whether we could find the space in a conference paper for such a large survey of results. As far as we understood the question, the suggested work would probably rather fit in a survey paper to be published in a journal.
>
> Finally, the reviewer asked for the performance of the D-TD3 and Q-TD3 ablations if MAP-Elites was used as a selection mechanism. As these ablations only use one criterion, using MAP-Elites for them does not make much sense. Nevertheless, we added the MAP-Elites version of our selection mechanism (noted -ME) in all benchmarks results for comparison with the version using a Pareto front.

---

> ### Author Response · Authors · 2020-11-24
> **Response to Reviewer4 (1/2)**
>
> The reviewer is asking for more references about QD methods early in the introduction. We agree that they were missing, we added several missing references.
>
> The reviewer is wondering why diversity is measured in terms of BDs and not directly in the parameter space or state space, and whether this is our own contribution in this paper. Actually, this contribution comes from (Doncieux et al., 2019) “Novelty search: a theoretical perspective”. Although it is written using the vocabulary of the evolutionary computation community rather than that of the machine learning community, this paper clearly explains that performing uniform search in a well chosen behavior descriptor space is the best one can do when facing a sparse reward problem, and it explains why this is hard. The reference to that paper was mistakenly lost during a refactoring of the introduction, we apologize, we have now fixed this and added it again.
> But, to answer a specific question of the reviewer, as far as we know, we are the first to combine QD approaches with RL gradient descent methods. This was mentioned in the related work section, we are now more explicit about this in the introduction of the paper, and more explicit about using this combination.
>
> The reviewer also asks if we are the first to use two critics. Actually, both SAC and TD3 use two critics and take the min over these critics to avoid overestimation bias, but this has nothing to do with our approach, where the two critics have a very different role.
> The reviewer is right in pointing that the way diversity is calculated was not properly explained in the "Problem Statement" section. Our point was to just explain state and solution Behavior Descriptor in the “Problem Statement”, and to defer the calculation of solution and state-related diversity in the Methods. This has been reformulated.
>
> The reviewer is asking why the "Deceptive Humanoid" was not evaluated in our paper, wondering whether the sample complexity reduction can be seen in this environment as well. We agree that providing comparisons between QD-RL and the ME-ES family of methods in a second benchmark would strengthen our conclusions. Unfortunately, the humanoid environment requires much larger computational resources than what we can afford. The ME-ES results on this environment were obtained with the large clusters of Uber Labs using 1000 CPUs. We tried but our experiments are still running after two weeks of computation. Conversely, we could not run the ME-ES family of methods in Ant-Trap, for the same reason. Nevertheless, we think it would be unfair to reject our paper based on the fact that we cannot afford these results in two weeks, as it would sound like restricting access to the conference to a small “club” of the big companies who can.
>
> The reviewer is also asking about a difference between the reported performance of several algorithms in Colas et al. 2020 and our work. Due to our limited computational budget already outlined above, we could not extensively tune the RL part of our algorithm and still get some instabilities for certain seeds. To compensate for that while still being fair in the comparison, we took the best 2 seeds over the 5 seeds we could run, and compared to the best 2 seeds over the results of Colas et al. 2020 (who provided their results so that we could perform the extraction). As a consequence, the reviewers can check that all the performances we report from the Colas et al. 2020 paper are increased with respect to the original work.

---

### Official Review · AnonReviewer1 · 2020-11-01
**Extremely well-written and compelling paper - reservations exist in terms of novelty, and uniformity of baselines**

**Rating:** 6
**Confidence:** 4

**Review:**

Summary: The paper introduces QD-RL, a population-based algorithm that combines off-policy RL with Quality-Diversity based techniques for continuous control problems. The core novelty is the decoupling and concurrent optimization of diversity and performance leveraging a population-based approach. Experiments in continuous control domains demonstrate that the proposed method is able to outperform prior evolutionary methods.

Technical Quality and Clarity: The paper is well written and easy to follow. An extensive review of related literature is included that is extremely helpful in situating the work amongst the broader field. The paper makes good use of visualizations and flowcharts to communicate the ideas clearly and succinctly. The proposed method also seems technically sound and coherent.

Novelty: The novelty of the proposed method seems marginal. Broadly, the proposed algorithm integrates diversity seeking optimizers based on archival behavior characterization (Quality Diversity / Map-Elites) within a hybrid framework that combine off-policy RL with an evolutionary population via a shared replay buffer (ERL / GEP-PG / CERL). The method then dedicates sub-populations within the evolutionary population towards co-optimizing diversity and performance - an idea similar to multiobjective speciation used widely in evolutionary literature.

Significance: I can see the value the proposed method could bring to the field by integrating ideas from multiple fronts towards a performant solution However, a number of questions remain towards appraising this value proposition:

Both mutation operators (diversity and performance seeking) used for QD-RL seem to be based on gradients computed from the off-policy RL update, unlike evolutionary methods that have some sense of randomness to it. While this assuredly would contribute in sample-efficiency, would the loss of a random mutation operator make it more likely to get stuck in local minima that greedy gradient-based methods are known to be prone to?

On the same note, how does the computational cost (related to backpropagation) scale with the population size as each individual’s parameter update relies on computing a policy gradient with subject to its parameters? Further, it also needs to access the same replay buffer to compute its update? How does this affect computational and memory complexity?
While tertiary to the idea, these are practical concerns essential to deployment in the real-world operation. Further, these are some key advantages of a gradient-free EA that the method is being compared to. Computing the diversity criterion using a nearest neighbour from a (large) archive also elicits a similar query.

The choice of baselines used to conduct the comparative study is very uniform. NS-ES, NSR-ES and NSRA-ES all originate in the same paper while ME-ES is a closely related method that builds on these very ideas. All these baselines are QD-methods that use ES as its optimizer. Without comparisons to a more representative sample of methods outside this family, it is difficult to quantify the significance of the results presented in the paper.
For example, the core differential in performance in the paper is in terms of sample efficiency (Table 2). However, the central technology that enables sample efficiency is the shared replay buffer that allows the gradient-based learner to leverage the same data (transition) multiple times across varying policy parameter distributions. This is a core tenet behind hybrid methods like ERL, GEP-PG, CERL, CEM-RL to name a few. At least one representative from this family should be included if the comparative claims hinges on sample efficiency.
Further, adding a general state-of-the-art technique for Mujoco-based continuous control like SAC, ARAC, PEARL would greatly strengthen the paper’s claims.

Overall, the paper presents an interesting method. My primary reservation lies in the marginal novelty of the idea, computational concerns, and the rather uniform set of baselines used. A satisfying response that addresses these concerns will influence my views on the paper.

########################### Post Rebuttal ##################
I have read the other reviews and the author's responses. I thank the authors for conducting the additional experiments and integrating the feedback from the reviews. Accordingly, I am raising my score.
Overall, I agree with the authors that combining QD with pg operators is novel - however, I am still not fully convinced that it is significant enough for a full paper at ICLR. This remains my primary reservation that prevented a higher score for the paper from my end.

---

> ### Author Response · Authors · 2020-11-24
> **Response to Reviewer1 (2/2)**
>
> The reviewer rightfully pointed out that using a shared replay buffer is crucial to the sample efficiency of QD-RL, thus QD-RL could be compared to other families of methods combining evolutionary and deep RL methods such as ERL, CERL, CEM-RL. We agree that this comparison was important and we added a CEM-RL baseline on both Ant-Trap and Ant-Maze environments. It happens that the key difference between QD-RL and CEM-RL (which must be representative of ERL and CERL too in that respect) is that CEM-RL lacks a diversity component, hence it gets stuck in traps and dead-ends, as any deep RL algorithm like TD3, despite improved exploration in the parameter space due to its CEM core.
>
> We also checked the importance of diversity in deep RL algorithms, as suggested by the reviewer, running SAC and TD3 (called Q-TD3 in our paper) on our benchmarks and found the same result. In particular, SAC on AntMaze failed to move the ant at all. The ant model in AntMaze is harder to control than that of Ant-Trap: in AntMaze, the ant barely moves at all when controlled by a random agent whereas in Gym Ant-v2, a random agent makes it jitter and move to random positions, resulting in easier bootstrapping of rewarded behaviors. Furthermore, the reward function in AntMaze does not contain the additional “control”, “healthy” and “contact” features, which are present in the reward function of Gym Ant-v2. We checked that our implementation of SAC obtains state-of-the-art results on standard Mujoco benchmarks such as Hopper, HalfCheetah, Humanoid and Ant, and we allocated appropriate time and computing resources to fine-tune it. Hence we assume that these poor results do not come from our implementation. In Ant-Trap, SAC, TD3 and CEM-RL do not manage to escape the trap, as they follow the deceptive gradient.
>
> We hope that these additional results will help convince the reviewer that the diversity component brought by QD-RL is crucial in our benchmarks, and is in itself a novel enough contribution in the field of deep RL.

---

> ### Author Response · Authors · 2020-11-24
> **Response to Reviewer1 (1/2)**
>
> First, the reviewer is questioning the novelty of the method. To our knowledge, our work is the first to combine QD methods with policy gradient descent operators from deep RL, resulting in a significantly better sample efficiency wrt standard QD methods, and better exploration capabilities when compared to deep RL methods. This was stated in the Related work section, we have insisted in the introduction.
>
> Then the reviewer asks about the potential higher propensity of purely gradient-based methods to get stuck into local optima with respect to evolutionary methods which have some randomness inside their improvement operator.
>
> The question is both important and tricky. The reviewer is right, it is well-known for instance that CMA-ES is performing an approximate form of Natural Gradient Descent (see Nikolaus Hansen’s papers) and that the corresponding “variation-selection” operators include some randomness, resulting in a better capability to escape small local minima than Batch Gradient Descent. However, several points have to be made. First, modern “adaptive” gradient descent techniques such as Adam include some momentum-like mechanisms that help them escape the same local minima. Second, Stochastic Gradient Descent (SGD) also benefits from some randomness (in the choice of the samples in mini batches) that has the same property. Second, we have to consider local minima at different scales. Pure novelty-seeking mechanisms help avoiding local minima of the reward function or ignoring deceptive gradients that would drive the policy training process far away from the optima we are looking for. Whereas small randomness or stochasticity can help avoid small minima during local policy improvement, these methods are generally not enough to escape the larger optima that we illustrate in our paper with navigation dead-ends or traps. Interestingly, CEM-RL which benefits from randomness but lacks a pure diversity component is still trapped in deceptive gradient problems.
>
> The reviewer is asking questions about :
> (a) how the computational cost scales with the population size.
> (b) the computational and memory impact of accessing a central replay buffer to compute updates.
> (c) the comparison between a gradient-free EA and QD-RL when considering deployment in the real-world operation.
>
> To answer all these questions, a short section about the parallel implementation of QD-RL has been added in Appendix B.4 (adding it in the main paper was not possible due to space restrictions). To answer more precisely here:
>
> (a) We parallelize our code and use 1 thread per individual in the population. The number of gradients computed scale linearly with the population size, but the runtime remains constant as the gradients are computed in parallel for each agent. We typically use 10-cores computers per run for our experiments making QD-RL very accessible compared to more expensive methods like IMPALA.
>
> (b) The current implementation of QD-RL centralizes the solution archive on a master worker to save memory and replicates the replay buffer among workers. During evaluation, each worker stores its data locally and the replay buffers are synchronised across all the threads at each iteration. We made this choice because the memory bottleneck is the archive and replay buffers contribute marginally to memory consumption. Currently, the largest experiments need about 10Gb RAM per run (with the reported hyperparameters), which we find reasonable considering the resources available nowadays. Memory consumption still scales linearly with the number of workers/agents but at a low rate, knowing that only replay buffers are replicated.
>
> (c) Following the above answers, QD-RL does not encounter any technical problem to scale up (scaling up meaning adding more agents in this case). Computing the diversity criterion using a nearest neighbour from a (large) archive is not a problem since 1) the archive size is limited and 2) very efficient tools (like “NearestNeighbors” from sklearn) make this calculation relatively quick.

---

### Official Review · AnonReviewer3 · 2020-11-02
**The significance and novelty of the paper's contributions is not sufficient.**

**Rating:** 3
**Confidence:** 5

**Review:**

The authors describe a QD-RL algorithm to solve continuous control problems with neural controllers. The authors state that  they maximize diversity within the population and ” the return of each individual agent”. Furthermore, the authors state that QD-RL selects agents from a Pareto front or from a Map-Elites grid. This paper is weak in estimating its performance in a clear way. The overall structure in Section 4 is not well defined and difficult to follow. Descriptions of the methods and technical details of the proposed study are incomplete. Furthermore, literature review simply lists studies without presenting a coherent and systematic introduction or critical evaluation. Overall, the contribution of the paper is not significant.

---

> ### Author Response · Authors · 2020-11-24
> **Response to Reviewer3**
>
> We thank the reviewer for taking the time to post a review but, honestly, there is not much we can extract from this review. The reviewer claims “The overall structure in Section 4 is not well defined and difficult to follow. Descriptions of the methods and technical details of the proposed study are incomplete. Furthermore, literature review simply lists studies without presenting a coherent and systematic introduction or critical evaluation.”
> Can the reviewer be more specific, point to concrete problems or missing technical details or papers? We are all the more surprised by these comments that the other reviewers, who have thoroughly read the paper, seem to strongly disagree.  Reviewer 1 says: “The paper is well written and easy to follow. An extensive review of related literature is included that is extremely helpful in situating the work amongst the broader field. The paper makes good use of visualizations and flowcharts to communicate the ideas clearly and succinctly. The proposed method also seems technically sound and coherent.” Reviewer 4 says: “The general flow of the paper is clear and is easy to follow. Related work gives a good overview of diversity-seeking approaches.”
> So we would appreciate getting more solid ground to understand the negative and “absolutely certain” opinion of the reviewer about our paper.

---

### Official Review · AnonReviewer2 · 2020-11-09
**Promising, but needs more experimental comparisons and better justification for the used mutation operator**

**Rating:** 6
**Confidence:** 3

**Review:**

Summary:

This paper is addressing the problem of hard exploration / escaping local minima in continuous control, by optimizing a population of agents for both environment reward and diversity, using both off-policy RL and Quality-Diversity (QD) optimization. Each agent is individually optimized using off-policy RL for either environment reward or diversity, and at a population level, individuals are selected using a QD method such as Map-Elites. On a hard exploration problem, Ant-Maze (Colas et al 2020), the proposed method is shown to be significantly more data-efficient compared to several other SOTA ES methods (ME-ES (Colas et al, 2020), NS-ES (Conti et al 2018), NSR-ES (Conti et al 2020)).

Reasons for score:

The Ant-Maze results are promising and the paper is clear and well written. However, there weren't enough comparisons to relevant baselines, while a number of important algorithmic choices weren't sufficiently justified and explored. I'll expand on these points in the following paragraphs.

The experiments on Ant-Maze are the only comparison of the proposed method (QD-RL) to other baselines meant to address difficult exploration problems. The inclusion of the following comparisons and experiments would make the paper much stronger:
(1) comparison of several baselines in several different hard-exploration environments (such as Point Maze and Ant-Trap used in ablation experiments),
(2) comparisons to PBT and RL with exploration bonuses.
I hope the authors will have such comparisons for the rebuttal phase, or if these comparisons are not suitable, I'd be interested to hear out their arguments.

Some important algorithmic choices in the population mutation part of the algorithm, do not seem well justified or sufficiently empirically explored:
(3) The use of a single quality and diversity critic for all agents in the population does not make much sense from the RL perspective. What was the reasoning there, why not maintain a separate critic for each agent?
(4) Why are the individual agents optimized for either quality or diversity, why not optimize for a mixture of both objectives? What determines whether an agent is optimized for quality vs diversity?
Separately:
(5) The choice of behavior descriptors (BD) is quite important. All the experiments in this paper involve one such function (agent xy coordinates and position increments). How sensitive is the performance wrt the choice of BDs? What about performance on non-navigation tasks?

Questions:
- to clarify, is the novelty score reward computed as the mean distance between the state's bd_t and the closest bd_t in entire training history, current episode or experience buffer?

Other comments:
- "Besides, evolutionary methods are very valuable when nothing is known about the function to optimize." --> can you be more precise?
- in Table 1 and 2, can you report the results and stds to two decimal points? Please also report the standard deviation for steps

-----------------------------------

UPDATE: The authors did a very good job at answering my questions and the new experimental results are very much welcomed, hence I'm updating my score from 4 to 6.

Given that this is a highly empirical paper with relatively little novelty in the key idea, more comparisons would be necessary to justify increasing the score further. While I sympathize with the lack of computational resources and access to implementations, taking some extra time to implement and run those comparisons can be done. The code for RL methods with exploration bonuses (e.g. pseudocounts, RND) is accessible and these methods are not too costly to run. Methods like PBT (whose results are typically reported using large computational resources), could be implemented and compared in a regime with much more constrained resources.

---

> ### Author Response · Authors · 2020-11-24
> **Response to Reviewer2 (2/2)**
>
> (5) The reviewer mentions that all experiments in this paper involve the same Behavioral Descriptors (BDs) and would like to know how sensitive the performance is wrt the choice of BDs. Additionally, the reviewer would like to know about performance on non-navigation tasks.
>
> We agree that our study is limited to locomotion/navigation tasks and always uses (x,y) coordinates of the final point as Behavioral Descriptors (BDs). Actually, this is the case of most Novelty Search and QD papers we are aware of. Besides, it is important to realize that, in the case of Ant-Trap, the reward function is computed as the ant forward velocity at time t and not as an Euclidean distance to a target, thus the BDs and the reward function are not directly aligned. Thus suggests that QD methods and our QD-RL approach could well be applied to different BD spaces, but we leave this study for future work. To avoid overclaiming about our contribution, we made an explicit reference to these navigation tasks in the introduction and would like to point out that most standard Mujoco tasks share the same properties.
>
> Additionally, the reviewer asks if the novelty score reward is computed as the mean distance between the state's bd_t and the closest bd_t over the entire training history, the current episode or the experience buffer. The novelty score reward is computed as the mean distance between the state BD and the closest state BDs over the entire training history. Encountered state BDs are preserved in a dedicated storage with insertion rules, thus the novelty score reward represents the novelty of a state BD relative to all previously encountered state BDs. To clarify this point, we added a paragraph in the Method section.
>
> The reviewer asked to be more precise about "Besides, evolutionary methods are very valuable when nothing is known about the function to optimize." We meant that they are the method of choice when gradient descent cannot be applied to the controller as this controller would itself be a black box, for instance if it is provided by a tier as an unknown parametric function. We agree that this is not so often the case. Actually, evolutionary methods are also valuable when the markov assumption does not hold (as the policy gradient theorem explicitly needs it to hold to be applied). We rephrased the sentence in the paper.
>
> Finally,  despite the query of the reviewer, we have opted not to include the two decimals for standard deviations in Table 2. The environment has a length of 70 on both axes and is considered solved when the distance from the agent to the goal is inferior to 10. Considering this, a two decimals precision does not provide  relevant  information.

---

> ### Author Response · Authors · 2020-11-24
> **Response to Reviewer2 (1/2)**
>
> We thank the reviewer for the thorough and valuable feedback.
>
> (1) The reviewer is asking for comparison of more baselines in hard-exploration environments such as Point Maze and Ant-Trap used in ablation experiments. We added these results, see Table 2 and Figure 3. Regarding the ME-ES family, though we think adding such results would improve the study, we could not run these algorithms as they require 1000 CPUs per run, as mentioned in the general response to all reviewers. We also doubt that the corresponding cost is worth the study, as we have good reasons to believe that what we observed in Ant-Maze in terms of sample efficiency should generalize to Ant-Trap.
>
> (2) The reviewer is asking for comparison with Population-based training and RL with exploration bonuses. In that respect, within the two weeks, we could only run CEM-RL and SAC, as these algorithms benefit from easily accessible implementations. We have shown that these algorithms together with the QD-TD3-SUM baseline all get stuck in a dead-end or trap in our experiments.
>
> (3) The reviewer points out that the use of shared quality and diversity critics for all agents in the population does not make much sense from the RL perspective and asks why not maintain a separate critic for each agent. Looking for a theoretical justification of this approach would be beyond the scope of our paper, which is mainly empirical. Here we slightly expand the version we have added in the paper to answer the reviewer’s point.
>
> Some reasons for sharing both critics are as follows. In evolutionary approaches, an agent may not be selected for a large number of generations. As a result, the corresponding critics can be outdated when the agent is selected again. A first consequence, as pointed out in Pourchot&Sigaud (2018), is that a local Quality critic would be performing under a too strong off-policy regime for algorithms like TD3 or SAC. That’s why CEM-RL also shares a unique critic over the population. In our context, an additional reason is that diversity is not stationary, as it depends on the current population. As a consequence, a local diversity critic would evaluate the agent based on an outdated picture of the evolving diversity. Actually, we tried the local critic solution and it failed. Moreover, as diversity is relative to the population and not to an individual, it is more natural to have it centralized. A side benefit is that both shared critics become accurate faster as they combine the experience of all agents.
>
> (4) The reviewer asks why are the individual agents optimized for either quality or diversity rather than a mixture of both objectives.
>
> Following this remark, we added a variant of QD-TD3 called QD-TD3-SUM, in which quality and diversity rewards are summed into a composite reward, and only one critic is used. Here we copy-paste the paragraphe we added in the methods to answer the Reviewer's point:
>
> We tested QD-TD3-SUM on ANT-MAZE and ANT-TRAP. QD-TD3-SUM manages to solve the task in ANT-MAZE but completely fails to learn anything in ANT-TRAP. Besides, QD-TD3-SUM requires more samples to solve ANT-MAZE. An issue is that the quality and diversity rewards may give rise to gradients that point to opposite directions. For instance, at the beginning of training in ANT-TRAP, the quality reward drives the ant forward whereas the diversity reward drives it backward so as to escape the trap to explore parts of the 2D space. Therefore, both rewards may often anneal each other, resulting in a zero sum and preventing any learning. This situation, while rarer, also happens in the ANT-MAZE environment and reduces sample efficiency.
>
> The reviewer also asks what determines whether an agent is optimized for quality vs diversity.
>
> As in Colas et al. (2020), during each iteration, we optimize half of the selected population for quality and half for diversity. The half is sampled uniformly among the population. An actor that has been optimized at a certain point for quality can be optimized some iterations later for diversity. We chose this 0.5 ratio not to bias the algorithm towards exploration or exploitation.

---

### Author Response · Authors · 2020-11-24
**General response to all reviewers**

We answer here the comments shared by several reviewers. Additional comments specific to each reviewer are answered in specific answers.

We sincerely thank all reviewers for their very useful feedback.

The comment that we received most often is that more experimental comparisons to baselines were needed. R1 was asking for comparisons to hybrid methods like ERL, GEP-PG, CERL, CEM-RL, for Mujoco-based continuous control like SAC, ARAC, PEARL. R2 was asking for more comparisons in Point Maze and Ant-Trap and comparisons to Population Based Training and RL with exploration bonuses. R4 was asking for results on Humanoid-Trap and more results with Map-Elites implementations of QD-RL.

In less than two weeks, we managed to perform many of these additional comparisons: we have included CEM-RL and SAC in our baselines, we have added an additional variant of QD-TD3, called QD-TD3-SUM, implementing a mixture of quality and diversity, we have compared all baselines in Ant-Trap and we have included Map-Elites implementations (noted -ME) in all benchmarks. Unfortunately, given our limited computational budget, our experiments on Humanoid-Trap are still running and too far from converging for publication. Humanoid is much more expensive than the rest. We believe that the Ant-Trap benchmark shows similar properties at a lower cost, and we think it would be unfair to reject our paper based on the fact that we cannot afford Humanoid-Trap results in two weeks, as it would sound like restricting access to the conference to a small “club” of the big companies who can (the ME-ES results on Humanoid-Trap were obtained with a large 1000 CPU cluster of Uber Labs). On the same line, we could not use the ME-ES family as baselines in Ant-Trap as these algorithms also require 1000 CPUs per run.

The discussion has been updated with all these additional results. Without any surprise, SAC and CEM-RL, which lack a diversity mechanism, get stuck in dead-ends. Interestingly, the QD-TD3-SUM variant also fails in Ant-Trap as the diversity and quality gradients cancel each other, while it just underperforms in Ant-Maze, where the gradient cancellation effect is weaker.

The other aspects are detailed in the response to each reviewer. By the way, a question of Reviewer 4 helped us realize that our sample efficiency gain over evolutionary methods was larger than we initially thought. See response to Reviewer 4 for details.

---

### Decision · Program_Chairs · 2021-01-07
**Final Decision**

**Decision:**

Reject

**Comment:**

Although originally all reviewers were leaning towards rejection, the authors have done a very good job at addressing their concerns, significantly strenghtening the paper. There is now a consensus towards weak acceptance, with the exception of R3. However, I have decided to ignore R3's review for the following reasons:
- The original review was way too short and uninformative
- R3 did not reply to the authors' request for more constructive feedback
- R3 did not reply to my own request (private email)

That being said, even if other reviewers decided to increase their score after the rebuttal and discussion period, none of them was particularly enthusiastic about it: this remains a borderline paper combining ideas that, although promising, are not particularly original. At this time it falls slightly short off meeting the bar for an ICLR publication. I do believe that combining ideas from the RL and evolutionary research communities is a promising research direction, and I encourage the authors to take into account the reviewers' remaining comments to polish their paper (in particular, adding even stronger empirical results, and ensuring the key take-aways are clearly communicated).